# HuNoV Non-Structural Protein P22 Induces Maturation of IL-1β and IL-18 and N-GSDMD-Dependent Pyroptosis through Activating NLRP3 Inflammasome

**DOI:** 10.3390/vaccines11050993

**Published:** 2023-05-17

**Authors:** Nini Chen, Peiyu Chen, Yanhe Zhou, Sidong Chen, Sitang Gong, Ming Fu, Lanlan Geng

**Affiliations:** 1Department of Gastroenterology, Guangzhou Women and Children’s Medical Center, Guangzhou Medical University, Guangzhou 510623, China2008690004@gzhmu.edu.cn (S.G.); 2State Key Laboratory of Virology, Wuhan Institute of Virology, Center for Biosafety Mega-Science, Chinese Academy of Sciences, Wuhan 430071, China

**Keywords:** HuNoV, P22, NLRP3 inflammasome, IL18, IL-1β, pyroptosis

## Abstract

Norovirus infection is the leading cause of foodborne gastroenteritis worldwide, causing more than 200,000 deaths each year. As a result of a lack of reproducible and robust in vitro culture systems and suitable animal models for human norovirus (HuNoV) infection, the pathogenesis of HuNoV is still poorly understood. In recent years, human intestinal enteroids (HIEs) have been successfully constructed and demonstrated to be able to support the replication of HuNoV. The NLRP3 inflammasome plays a key role in host innate immune responses by activating caspase1 to facilitate IL-1β and IL-18 secretion and N-GSDMD-driven apoptosis, while NLRP3 inflammasome overactivation plays an important role in the development of various inflammatory diseases. Here, we found that HuNoV activated enteric stem cell-derived human intestinal enteroids (HIEs) NLRP3 inflammasome, which was confirmed by transfection of Caco2 cells with full-length cDNA clones of HuNoV. Further, we found that HuNoV non-structural protein P22 activated the NLRP3 inflammasome and then matured IL-1β and IL-18 and processed the cleavage of gasdermin-D (GSDMD) to N-GSDMD, leading to pyroptosis. Besides, berberine (BBR) could ameliorate the pyroptosis caused by HuNoV and P22 by inhibiting NLRP3 inflammasome activation. Together, these results reveal new insights into the mechanisms of inflammation and cell death caused by HuNoV and provide potential treatments.

## 1. Introduction

Norovirus can be transmitted in various ways, including direct transmission through fecal-oral and inhalation of aerosolized vomit and indirect transmission through contaminated water, food, and surfaces. Besides, the virus is highly contagious, and it takes as little as 18 virus particles to establish infection [1]. Norovirus remains infectious on surfaces for up to 2 weeks (more than 2 months in water) and is resistant to many common disinfectants. Outbreaks of acute gastroenteritis typically occur in crowded environments, including hospitals, schools, and cruise ships, although the most common route is believed to be through infected food handlers or contact with fecal-contaminated water [2]. Because of the highly contagious and efficient transmission of norovirus, emerging virus strains have the potential to cause a global pandemic. Globally, norovirus causes approximately 0.7 billion illnesses and 0.22 million deaths per year, with direct and indirect medical costs of more than $4 billion and $60 billion, respectively [3,4,5]. The lack of reproducible and robust cell culture systems and suitable animal models represents a major challenge in studying the pathogenesis of norovirus gastroenteritis and the effects of prevention and treatment measures [6,7]. Human intestinal enteroids (HIEs) can successfully cultivate HuNoV and are selected as a useful model [8,9,10]. However, the scarcity of human intestinal tissues discarded clinically and the complex construction process of HIEs limit their wider application.

HuNoV is a non-enveloped single-stranded positive RNA virus that belongs to the caliciviridae family. The diameter of the virion is about 27 to 40 nm, and the approximately 7.5 kb genome contains three open reading frames (ORFs), such as ORF1, ORF2, and ORF3. A non-structural polyprotein is encoded by ORF1 and then cleaved by its own protease to produce at least six different non-structural proteins involved in viral replication. ORF2 encodes the primary capsid protein VP1, and ORF3 encodes the secondary capsid protein VP2 [11]. A total of 180 copies of VP1 and a few copies of VP2 compose the capsid of the virus, and in vitro expression of the VP1 gene spontaneously forms a virus-like particle (VLP). HuNoV is a major cause of sporadic episodes of acute diarrhea and acute gastroenteritis outbreaks in all age groups [12,13]. Noroviruses are officially divided into six genogroups (GI-GVI), and they can be further divided into ten genogroups (GI-GX) based on VP1 amino acid sequence diversity, which can be further divided into forty-nine genotypes based on amino acid sequences of the VP1 [14]. Currently, the GII.4 genotype causes the majority of gastroenteritis outbreaks, although other genotypes, such as GII.17-caused cases, are on the rise [8,15]. In most cases, norovirus symptoms are mild and self-limited, but the elderly and immunocompromised may have serious and long-term symptoms [16,17], indicating that investigating the pathogenesis of norovirus is very important. While a recent study showed that murine norovirus (MNV) infection could induce the activation of the NLRP3 inflammasome and pyroptosis driven by gasdermin D (GSDMD) [18], no studies had shown that HuNoV shared the same features.

The NLRP3 inflammasome is assembled by NLRP3, ASC, and caspase1, and it is critical for host immune defenses against pathogen invasion, such as bacterial, fungal, and viral infections, and homeostasis. NLRP3 is mainly composed of three domains, including a leucine-rich repeat domain at the C-terminal, a pyrin domain at the amino-terminal, and a central nucleotide-binding and oligomerization domain (also known as NACHT). ASC is mainly composed of a caspase recruitment domain (CARD) at the C-terminal and a pyrin domain at the amino-terminal. The full-length caspase is mainly composed of three parts, including a caspase recruitment domain (CARD), a large catalytic domain, and a small catalytic domain. As a sensor, NLRP3 responds to various stimuli (danger-associated molecular patterns (DAMPs) and pathogen-associated molecular patterns (PAMPs), and undergoes self-oligomerization through interactions between their NACHT domains. The pyrin domains of oligomeric NLRP3 and ASC bind to each other, inducing the aggregation of ASC into a ASC speck. Subsequently, the ASC speck recruits pro-caspase-1 through interactions between their CARD domains to form the protein complex of NLRP3-ASC-pro-caspase1, leading to proximity-induced caspase-1 oligomerization and autocatalysis and then the release of the active fragments. The NLRP3 inflammasome is essential for host immune defense against pathogen invasion, such as bacterial, fungal, and viral infections, and homeostasis, but when the regulation is disordered, it contributes to the pathogenesis of a variety of inflammatory disorders, such as gout, cryopyrin-associated periodic syndromes (CAPS), diabetes, Alzheimer’s disease, and atherosclerosis [19,20,21]. When the NLRP3 inflammasome is activated, caspase1 is self-cleaved and activated, resulting in the maturation and secretion of IL-1β and IL-18, which are pro-inflammatory cytokines [22,23]. Beyond that, GSDMD could be cleaved by activated caspase1, and its N-terminal domain (N-GSDMD) is then released [24]. Further, N-GSDMD migrates to the cell membrane and binds to its extracellular receptors, interfering with the membrane structure and forming pores, thereby releasing various cell contents, IL-1β and IL-18 included, then finally activating a strong inflammatory response and causing pyroptosis, which is one of the programmed cell death modes [25,26,27]. Therefore, the activation of the NLRP3 inflammasome requires fine regulation, and its comprehensive mechanism needs to be fully investigated.

As a natural isoquinoline alkaloid, berberine (BBR) can be isolated from several traditional Chinese herbal plants, including *Coptis chinensis*, *B. petiolaris*, *Berberis aristate*, and *B. vulgaris* [28], and it shows great antiviral and antibacterial effects with few side effects [29]. As reported, BBR inhibits the priming of the NLRP3 inflammasome by decreasing the expression of NLRP3 by suppressing the activation of the nuclear transcription factor-kappa B (NF-κB) signaling pathway and inhibits the assembly of the NLRP3 inflammasome by affecting the binding between caspase1 and ASC [30,31], suggesting that BBR could suppress the activation of the NLRP3 inflammasome. Whether BBR has a potential role in suppressing the activation of the NLRP3 inflammasome caused by HuNoV has not been reported.

Here we revealed that HuNoV could activate the NLRP3 inflammasome, which further led to the maturation and release of the pro-inflammatory cytokines IL-1β and IL-18 and pyroptosis driven by N-GSDMD. In addition, we further revealed that HuNoV non-structural protein P22 contributes to the maturation and release of the pro-inflammatory cytokines IL-1β and IL-18 and N-GSDMD-dependent pyroptosis. Moreover, the NLRP3 inflammasome activation induced by HuNoV and P22 could be inhibited by treatment with BBR, suggesting that BBR has the potential to be a therapeutic agent for relieving norovirus gastroenteritis.

## 2. Materials and Methods

### 2.1. Viruses, Cell Culture, and Antibodies

The sequence of HuNoV GII.4 used here (Gene bank: OL721917) was isolated from positive stool samples of diarrhea patients [8]. Briefly, QIAamp RNA Blood Mini Kit (Qiagen, 52304, Hilden, Germany) was used to isolate the HuNoV genome from the positive stool sample, and then moloney murine leukemia virus (M-MLV) reverse transcriptase (Promega, M170B, Dane County, WI, USA) was used to synthesize the HuNoV cDNA. The progeny viruses were obtained by transfecting HEK293T cells with the plasmid, which encoded the full-length cDNA of HuNoV. Virus stocks were aliquoted and stored at −80 °C until use, and the genome copy was determined. Human colon epithelial cell lines Caco2 cells and HEK293T cells were purchased from ATCC and cultured in Dulbecco’s modified eagle’s medium (DMEM) containing 10% fetal bovine serum (FBS), purchased from Thermo Scientific, Sydney, Australia, and 100 U/mL penicillin/streptomycin (Genom, Hangzhou, China) at 37 °C with 5% CO2 in an incubator. Antibodies against NLRP3 (19771-1-AP), caspase1 (22915-1-AP), N-GSDMD (20770-1-AP), and β-tubulin (10068-1-AP) were purchased from Proteintech, Wuhan, China. Besides, HRP-conjugated goat anti-rabbit IgG (AS063) or goat anti-mouse IgG (AS003) were also purchased from Abclonal, Wuhan, China.

### 2.2. HIEs (Human Intestinal Enteroids) Construction

As mentioned previously, the construction of HIEs was performed [8,9]. In brief, discarded human intestinal tissues were obtained from the Department of Gastroenterology, Guangzhou Women and Children’s Medical Center, to isolate human crypts. We recommend removing as much fat from the tissue as possible and cutting them into small pieces of 3–5 mm. After immersing those tissues at 4 °C overnight, these crypts were isolated into the cold chelation solution (CCS) (8 mmol/L KH2PO4, 96.2 mmol/L NaCl, 0.5 mmol/L DL-dithiothreitol, and 2 mmol/L EDTA, 5.6 mmol/L Na2HPO4, 43.4 mmol/L sucrose, 54.9 mmol/L D-sorbitol, 1.6 mmol/L KCl). After further digestion and collection, these crypts were counted and used to construct three-dimensional (3D) culture of HIEs. The 3D culture model was constructed by Matrigel (CORNING, 354230, Corning, NY, USA), and complete medium CMGF+, containing 20% R-spondin 1, 10% Noggin-Fc, 50% L-WNT3A, 1 × N2 (Invitrogen, 15502-048, Waltham, MA, USA), 1 mM N-acetylcysteine (Sigma-Aldrich, A0737, St. Louis, MO, USA), 1 × B27 (Invitrogen, 17504-044, USA), 10 mM nicotinamide (Sigma-Aldrich, N0636, USA), 500 nM A-83-01 (Tocris, 2939, Shanghai, China), 50 ng/mL EGF (Invitrogen, PMG8044, USA), 10 µM Y-27632 (MedChemExpress, HY-10071, Shanghai, China), 10 µM SB202190 (Sigma-Aldrich, S7067, USA), and 10 nM Gastrin I (Sigma-Aldrich, 05-23-2301, USA), was used to culture the cell model. The medium was updated with CMGF+ medium every two days until HIEs were ready for passage. For differentiation, culture medium was replaced with differentiation medium, which contained the same components as CMGF+ medium apart from L-WNT3A, nicotinamide, R-Spondin1, Y-27632, and SB202190. Additionally, Noggin-Fc was also reduced by 50%. The differentiation medium was updated every two days before the HIEs were ready for further experiments.

After 7 days of differentiation, HIEs were released from the 3D-culture model and seeded overnight in 6-well plates for use the next day. They were then infected with HuNoV at a genome copy of 3.0 × 10^7^ and incubated at 37 °C with 5% CO_2_. Forty-eight hours later, the culture supernatants were collected for later use.

### 2.3. Plasmid Constructs

The clone encoding the full-length cDNA of HuNoV was obtained as described previously [8], and then we used the clone as a template to amplify the gene of HuNoV non-structural protein P22 with a 3 × Flag tag introduced into its C-terminal, and then the PCR products were inserted into the vector pcDNA3.1(+) (Invitrogen, Waltham, MA, USA), named 3.1-P22.

### 2.4. Cell Transfection, siRNA Interference, and Chemical Treatment

Caco2 cells were seeded in a 6-well plate one night in advance and then transfected with plasmids expressing the full-length HuNoV cDNA or P22 or empty vector pcDNA3.1(+) (negative control (NC)) using Lipo8000^TM^ (Beyotime, Shanghai, China) according to the manufacturer’s instructions. Briefly, for each well, 125 µL of Opti-MEM^®^ Medium was added into a 1.5 mL centrifuge tube, then 2 µg of plasmid was added, and 3.2 µL of lipo8000 was added after gentle blowing and mixing, and then the mixture was added into the cells and incubated at 37 °C with 5% CO_2_. Four to six hours later, the culture medium was replaced with fresh complete medium, and the incubation was continued for 48 h at 37 °C with 5% CO_2_. In the siRNA interference experiment, Caco2 cells were transfected with caspase1 or NLRP3-specific siRNA or negative control siRNA (si-NC) (Ruibo, Guangzhou, China) using Lipo8000^TM^ according to the manufacturer’s instructions for 12 h and then transfected with plasmids expressing the full-length cDNA of HuNoV or empty vector pcDNA3.1(+) (negative control (NC)). After incubation for four to six hours at 37 °C with 5% CO_2_, the culture medium was replaced with fresh complete medium, and the incubation was continued for 48 h at 37 °C with 5% CO_2_. In BBR treatment experiment, Caco2 cells were transfected with plasmids expressing the full-length cDNA of HuNoV or plasmids expressing P22 or empty vector pcDNA3.1(+) (negative control (NC)) and incubated for 24 h at 37 °C with 5% CO_2_. Then the transfected cells were treated with BBR (100 μM, Solarbio, Guangzhou, China) for 48 h at 37 °C with 5% CO_2_. Before being treated with BBR, the medium was replaced with fresh, complete medium. Finally, the cells and supernatants were separately collected for later use.

### 2.5. Enzyme-Linked Immunosorbent Assay (ELISA)

We collected the cell culture supernatants and then stored them at −80 °C until use. The concentrations of IL-1β and IL-18 in the supernatants of cell culture were determined by the Human IL-1β ELISA kit (MEIMIAN, MM-0181H2, Jiangsu, China) and the Human IL-18 ELISA kit (JINMEI, JM-03294H1, Jiangsu, China) according to the manufacturer’s instructions. In brief, a volume of 50 μL of gradient-diluted standards and samples was separately added to each well, and all of them were conducted in triplicate. After incubating at 37 °C for 30–60 min, the plates were washed 5 times with 1× washing buffer. Then HRP-conjugated reagent in a volume of 100 μL was added to each well and incubated at 37 °C for 30–60 min. The plates were then washed 5 times again with 1 × washing buffer, and chromogen A 50 μL and chromogen B 50 μL were added, and the plates were incubated for 10 min at 37 °C. Finally, the action was stopped by adding stop solution (2 mol/L H_2_SO_4_) in a volume of 50 μL to each well. The microplate reader (Molecular Devices, Silicon Valley, USA) was used, and the signal was quantified at 450 nm wavelength.

### 2.6. Protein Concentration Detection

The total protein concentration was determined by the BCA protein concentration determination kit (Beyotime, P0010, Shanghai, China). BCA reagent A and BCA reagent B were thoroughly mixed in a ratio of 50:1 to form the BCA working solution. Gradiently diluted standards and cell lysates in a volume of 20 μL was separately added to each well. After incubating at room temperature for 60 min, 200 μL of BCA working liquid was added to each well. All standards and samples were added in duplicate to the 96-well plate. The microplate reader (Molecular Devices, USA) was used, and 562 nm wavelength or wavelengths between 540 and 595 nm were set to quantify the signal. Ultimately, we calculated the total protein concentration of the sample based on the standard curve.

### 2.7. Western Blotting Analysis

These experiments were performed as mentioned previously [32]. In brief, the collected cells were lysed on ice for 0.5–1 h with cell lysis (Beyotime, P0013, Shanghai, China) with protease inhibitor cocktail (Roche, 11697498001, Hamburg, Germany). Then the lysate was collected into a 1.5 mL centrifuge tube and centrifuged at 10000~15000 rpm for 12 min at 4 °C to obtain the supernatant. The protein lysate was determined for protein concentration. According to the manufacturer’s instructions, the total protein concentration was determined by BCA assay kit (ThermoFisher, BCA-23227, Waltham, MA, USA), and then we boiled the samples with loading buffer (50 mM Tris-HCl, 25% glycerol, 2% SDS, 1% DTT, pH 6.8) for 10 min. Further, the prepared samples were subjected to 10% SDS-PAGE and transferred onto 0.45 µm PVDF membranes (Merck Millipore, BS-00-2529, Darmstadt, Germany). We recommend soaking the PVDF membrane in methanol for at least 10 s before use. An amount of 5% non-fat milk blocked the above PVDF membranes at room temperature for 60 min, and subsequently, the primary antibodies against NLRP3, caspase1, N-GSDMD, and β-tubulin incubated them overnight at 4 °C or 60 min at room temperature. These antibodies were purchased from Proteintech, Wuhan, China. After being washed with TBST (200 mM NaCl, 50 mM Tris-HCl, 0.1% Tween-20) for 3 times, 5 min each time, the membrane was incubated with HRP-conjugated goat anti-mouse IgG (Proteintech, SA00001-1, China) or HRP-conjugated goat anti-rabbit IgG (Proteintech, B900210, China) at room temperature for 60 min. After washing with TBST for 5 times, 5 min each time, we incubated the PVDF membranes with enhanced chemiluminescence (ECL) (Biosharp, BL523B, Wuhan, China) and visualized the protein bands under a chemiluminescent imaging system. Image J was used to quantify the relative intensities of the western blots.

### 2.8. Cell Viability Assay

CCK-8 counting kit (Zeta life, K009) was used to measure the cell viability. CCK-8 solution contains WST-8 (2-(2-methoxy-4-nitrophenyl)-3-(4-nitrophenyl)-5-(2, 4-disulfophenyl)-2H-tetrazolium, monosodium salt), which can be reduced by dehydrogenases in mitochondria to form a highly water-soluble orange formazan, directly proportional to cell proliferation and inversely proportional to cell toxicity. Following the manufacturer’s instructions, Caco2 cells at a density of 2.5 × 10^3^ cells in 100 μL of medium were plated into each well of a 96-well plate. After transfecting Caco2 cells for 48 or 72 h, CCK-8 solution was added to each well in a volume of 10 μL and incubated for 90 min at 37 °C. The microplate reader (Molecular Devices, USA) was used, and the absorbance was determined at 450 nm wavelength. Control group was assigned 100% viability, and cell viability = (treatment group/NC group) × 100%.

### 2.9. Statistical Analysis

All data were expressed as mean ± SD unless otherwise indicated. GraphPad Prism 8.0 (San Diego, CA, USA) was used to perform statistical analysis. Analyzing the data to determine the differences between two groups was conducted under the Student’s *t* test. One-way ANOVA and Student-Newman-Keuls (SNK) post hoc were used to analyze the data to determine the difference among three or more groups. * *p* < 0.05 was considered statistically significant.

## 3. Results

### 3.1. HuNoV Increases Proteolytic Maturation of IL-1β and IL-18

As a result of the lack of reproducible and robust in vitro culture systems and suitable animal models for HuNoV infection, the pathogenesis of HuNoV is still poorly understood. As described previously [8], HuNoV could infect HIEs. The HIE culture system was used to assess the effect of HuNoV on cytokines associated with inflammation, and we found that the secretion of pro-inflammatory cytokines IL-1β and IL-18 was significantly increased (Figure 1A). We obtained the same results by transfecting Caco2 cells with plasmids encoding the full-length cDNA of HuNoV (Figure 1B). The above results suggested that HuNoV could increase the maturation and secretion of IL-1β and IL-18.

### 3.2. HuNoV Increases Secretion of IL-1β and IL-18 by Activating NLRP3 Inflammasome

The synthesis and release of IL-1β and IL-18 require two types of signaling, one promoting their expression and the other promoting their maturation and release by activating the NLRP3 inflammasome. Due to the scarcity of human intestinal tissues discarded clinically and the complexity of the construction process of HIEs, Caco2 cells were used as models for later research. To investigate whether HuNoV increased the secretion of IL-1β and IL-18 through activating the NLRP3 inflammasome, Caco2 cells were transfected with the plasmids encoding full-length HuNoV cDNA or empty vector (NC), and we found that the expressions of NLRP3, caspase1, and cleaved caspase1 in Caco2 cells were significantly up-regulated (Figure 2A,B). While co-transfecting Caco2 cells with specifically NLRP3 or caspase-1-targeted siRNA, the maturation of caspase-1 was obviously inhibited, and the secretion of IL-1β and IL-18 was significantly inhibited, as shown in Figure 2C–E. The above results indicated that NLRP3 inflammasome activation by HuNoV provoked the maturation and secretion of IL-1β and IL-18 in enterocytes.

### 3.3. HuNoV Induces N-GSDMD-Dependent Pyroptosis by Activating NLRP3 Inflammasome

Inflammasomes are known to provoke a lytic cell death mode termed pyroptosis, which is provoked by the proteolytic processing of GSDMD to N-GSDMD. NLRP3 inflammasome activation could activate caspase-1 and then cleave GSDMD and release N-GSDMD, leading to N-GSDMD-driven pyroptosis [25,27]. To investigate whether the activation of the NLRP3 inflammasome induced by HuNoV increased the expression of N-GSDMD, we transfected Caco2 cells with the plasmid encoding a full-length HuNoV cDNA clone and found that the level of N-GSDMD was obviously increased (Figure 3A,B), as was the increased cell death (Figure 3C). While the increased N-GSDMD level and pyroptosis were inhibited when co-transfected with specifically NLRP3 or caspase1 targeted siRNA (Figure 3D–F), indicating that N-GSDMD-dependent pyroptosis caused by HuNoV is dependent on the activation of the NLRP3 inflammasome.

### 3.4. P22 Induces Maturation of IL-1β and IL-18 and N-GSDMD-Dependent Pyroptosis by Activating NLRP3 Inflammasome in Caco2 Cells

P22 is a non-structural protein encoded by HuNoV [33,34], and we transfected Caco2 cells with a P22-encoding plasmid or empty vector (NC) and found that P22 significantly increased secretion of IL-1β and IL-18 (Figure 4A). Further, we found that P22 also increased the expression of NLRP3 and cleaved-caspase1, as well as the proteolytic processing of GSDMD to N-GSDMD (Figure 4B,C), and we also found that P22-specific siRNA inhibited the full-length HuNoV cDNA clone that activated the NLRP3 inflammasome (Figure 4D), suggesting that P22 plays an important role in the activation of NLRP3 inflammasome and N-GSDMD-driven pyroptosis caused by HuNoV.

### 3.5. BBR Inhibits the Activation of NLRP3 Inflammasome and N-GSDMD-Driven Pyroptosis Induced by HuNoV and P22

Berberine (BBR) is traditionally used to treat diarrhea and gastroenteritis and has been reported to inhibit NLRP3 inflammasome activation [30,35,36]. To explore whether BBR could inhibit HuNoV or P22-induced activation of the NLRP3 inflammasome and N-GSDMD-driven pyroptosis, we transfected Caco2 cells with plasmids encoding the full-length cDNA of HuNoV or P22, and an empty vector was used as a negative control (NC). As shown in Figure 5A, BBR could rescue the cytotoxicity caused by HuNoV and P22. Besides, HuNoV and P22 increased cleaved capase1, N-GSDMD, and secreted IL-1β and IL-18, while BBR inhibited the increase (Figure 5B–D), indicating that BBR could inhibit the activation of the NLRP3 inflammasome and N-GSDMD-dependent pyroptosis induced by HuNoV and P22 and suggesting that BBR has the potential to treat the acute gastroenteritis caused by HuNoV infection.

## 4. Discussion

Norovirus infection is a major cause of gastroenteritis, and outbreaks are frequent, resulting in a serious medical burden. People of all ages, especially infants and young children, the elderly, and immunocompromised patients, tend to be infected by HuNoV. Several factors are currently increasing the global health challenge of norovirus infection, in particular the increasing number of infected people who are immunocompromised. In addition, the rapid evolution of circulating norovirus genes and antigens complicates the development of vaccines and therapies, which urgently needs to be resolved. While the pathogenesis of HuNoV remains unclear because of the lack of reproducible and robust in vitro culture systems and suitable animal models of infection, currently, human intestinal enteroids (HIEs), which support the infection and replication of HuNoV, have been successfully constructed and have been widely used in human norovirus research. However, the lack of human intestinal tissues discarded clinically and the complexity of the construction process of HIEs limit their wider application. In this study, we constructed a 3D culture model of HIEs and found that HuNoV-infected HIEs showed increased pro-inflammatory cytokines IL-1β and IL-18 in the cell supernatants, and the full-length HuNoV cDNA clone transfected Caco2 cells showed the same results, which were consistent with the phenomenon that acute gastroenteritis is easily caused by HuNoV infection. When the NLRP3 inflammasome, assembled by NLRP3, ASC, and caspase1, is activated, caspase1 is self-cleaved and activated, resulting in the maturation and release of IL-1β and IL-18 [19,20]. We further investigated the activation of the NLRP3 inflammasome and found that HuNoV increased the expression of NLRP3, caspase1, and cleaved-caspase1, while down-regulating NLRP3 or caspase1 inhibited the secretion of IL-1β and IL-18, which indicated that HuNoV promoted the maturation and secretion of IL-1β and IL-18 through activating the NLRP3 inflammasome. NLRP3 inflammasome activation could also cleave GSDMD and release N-GSDMD. N-GSDMD can migrate to the cell membrane and bind to its extracellular receptors, interfering with membrane structure and forming pores, thereby releasing various cell contents, IL-1β and IL-18 included, then activating a violent inflammatory response and causing pyroptosis, which is one of the programmed cell death modes [25]. We investigated the expression of N-GSDMD and cell death, and the results showed that HuNoV increased N-GSDMD expression as well as cell death, which were inhibited by down-regulating NLRP3 or caspase1 by specific siRNA targeting NLRP3 or caspase1. The above results suggest that HuNoV could promote the maturation and release of IL-1β and IL-18, as well as N-GSDMD-driven pyroptosis, depending on the activation of the NLRP3 inflammasome. We further revealed that HuNoV non-structural protein P22 played an important role in activating the NLRP3 inflammasome and N-GSDMD-driven pyroptosis, which suggested that P22 might be a potential therapeutic target. It is worth noting that overexpression of P22 may not completely reflect what happens in HuNoV-infected cells, and whether other HuNoV proteins influence P22’s regulation of NLRP3 inflammasome activation needs to be further studied.

Berberine (BBR), a natural isoquinoline alkaloid isolated from several traditional Chinese herbal plants, is traditionally used to treat diarrhea and gastroenteritis [28,37], and it was reported to inhibit the activation of the NLRP3 inflammasome [38]. Here, we intended to investigate whether BBR could inhibit NLRP3 inflammasome activation caused by HuNoV and found that BBR treatment reduced the secretion of IL-1β and IL-18 and suppressed N-GSDMD-driven pyroptosis through inhibiting the activation of NLRP3 inflammasome caused by HuNoV and P22, suggesting that BBR has the potential to treat HuNoV-induced acute gastroenteritis through inhibiting the inflammatory response and pyroptosis.

## 5. Conclusions

In conclusion, we found that HuNoV activated the NLRP3 inflammasome, leading to the maturation and release of IL-1β and IL-18, as well as N-GSDMD-driven pyroptosis. Our work will contribute to a better understanding of HuNoV-induced inflammatory and cell death pathways. Besides, we identified that BBR treatment inhibited NLRP3 inflammasome activation induced by HuNoV and HuNoV non-structural protein P22, which suggested the potential role of BBR in curing HuNoV-caused acute gastroenteritis.

## Figures and Tables

**Figure 1 vaccines-11-00993-f001:**
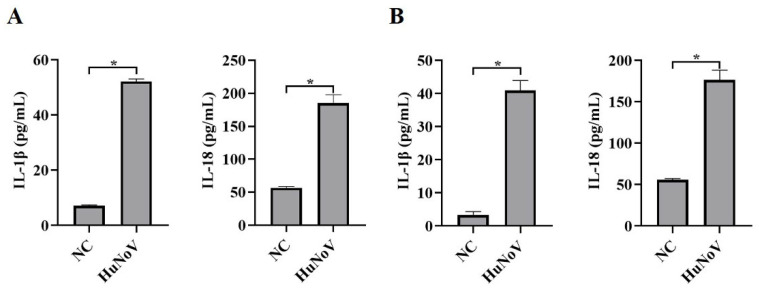
HuNoV facilitates the secretion of IL-1β and IL-18. (**A**) HIEs were infected with HuNoV at a genome copy of 3.0 × 10^7^ or treated with the virus preservation solution (NC) for 48 h, and the culture supernatants were collected for analyzing secreted IL-1β and IL-18 levels by ELISA. (**B**) Caco2 cells were transfected with the full-length HuNoV cDNA clone or empty vector (NC) for 48 h, and the culture supernatants were collected for analyzing secreted IL-1β and IL-18 levels by ELISA. * *p* < 0.05.

**Figure 2 vaccines-11-00993-f002:**
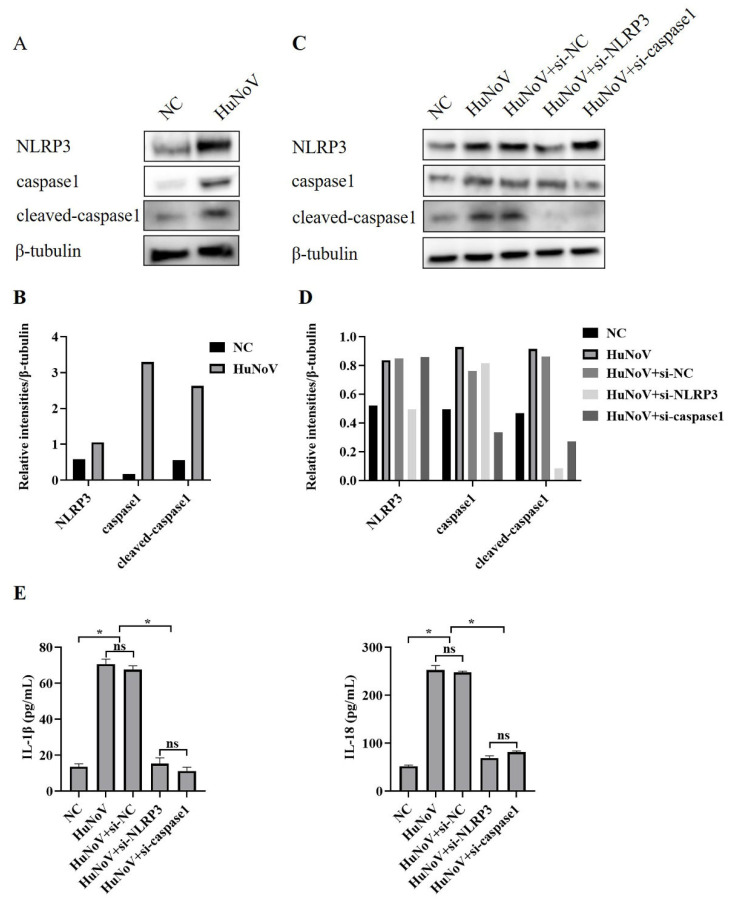
NLRP3 activation in Caco2 cells induced by HuNoV. (**A**,**B**) Caco2 cells transfected with plasmids encoding the full-length cDNA of HuNoV or empty vector (NC) for 48 h were lysed, then immunoblotted for NLRP3, caspase1, cleaved-caspasse1, and β-tubulin, and these blots were quantified by Image J. Caco2 cells were transfected with siRNA (si-NC, si-NLRP3, and si-caspase1) for 12 h, and then the cells were transfected with empty vector (NC) or the plasmids encoding the full-length cDNA of HuNoV. After 48 h of culture, we collected the cell lysates and supernatants. (**C**,**D**) The expressions of NLRP3, caspase1, cleaved-caspase1, and β-tubulin in cell lysates were analyzed by Western blot and quantified by Image J. (**E**) The levels of secreted IL-1β and IL-18 in cell culture supernatants were measured by ELISA. * *p* < 0.05; ns, no significant difference.

**Figure 3 vaccines-11-00993-f003:**
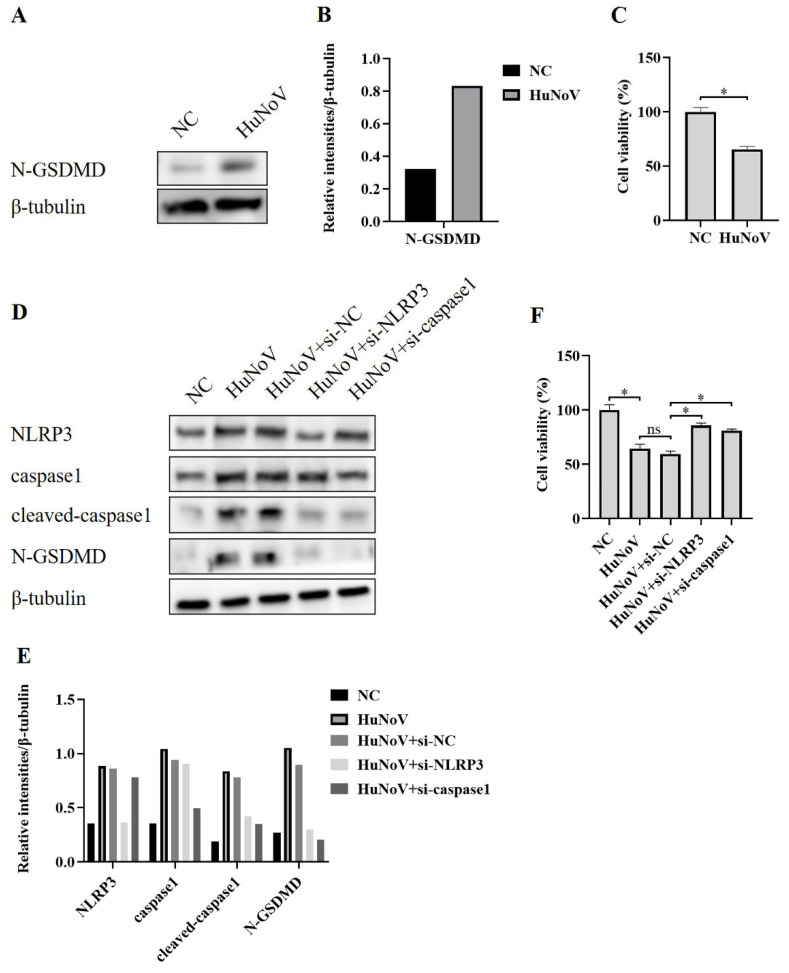
NLRP3 inflammasome activation caused by HuNoV-induced N-GSDMD-driven pyroptosis. Caco2 cells were transfected with the full-length HuNoV cDNA clone or empty vector as negative control (NC) for 48 h. (**A**,**B**) Cells were lysed and then immunoblotted for N-GSDMD and β-tubulin, and these blots were quantified by Image J. (**C**) CCK-8 assay detected the cell viability of Caco2 cells. Transfecting Caco2 cells with siRNA (si-NC, si-NLRP3, and si-caspase1) for 12 h, and then the cells were transfected with empty vector as negative control (NC) or plasmids encoding the full-length cDNA of HuNoV for 48 h. (**D**,**E**) Cells were lysed and then immunoblotted for NLRP3, caspase1, cleaved-caspasse1, N-GSDMD, and β-tubulin, and these blots were quantified by Image J. (**F**) The cell viability of Caco2 cells was detected by CCK-8 assay. * *p* < 0.05; ns, no significant difference.

**Figure 4 vaccines-11-00993-f004:**
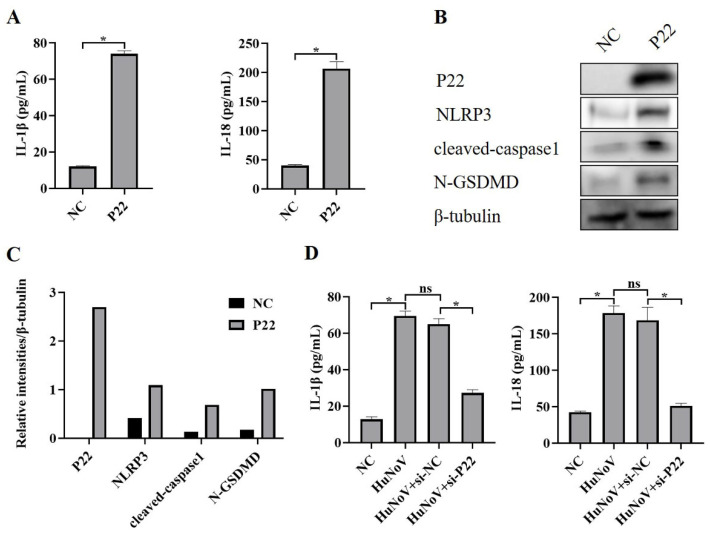
P22 activates NLRP3 inflammasome and N-GSDMD-dependent pyroptosis. Caco2 cells were transfected with P22 or empty vector (NC). (**A**) The levels of secreted IL-1β and IL-18 in cell culture supernatants were collected and measured by ELISA. (**B**,**C**) The expressions of P22, NLRP3, cleaved-caspase1, N-GSDMD, and β-tubulin in cell lysates were analyzed by Western blot, and then these blots were quantified by Image J. Transfecting Caco2 cells with P22-specific siRNA (si-P22) or negative control siRNA (si-NC) for 12 h, and then the cells were transfected with empty vector as negative control (NC) or plasmids encoding the full-length cDNA of HuNoV for 48 h. (**D**) The levels of secreted IL-1β and IL-18 in cell culture supernatants were collected and measured by ELISA. * *p* < 0.05; ns, no significant difference.

**Figure 5 vaccines-11-00993-f005:**
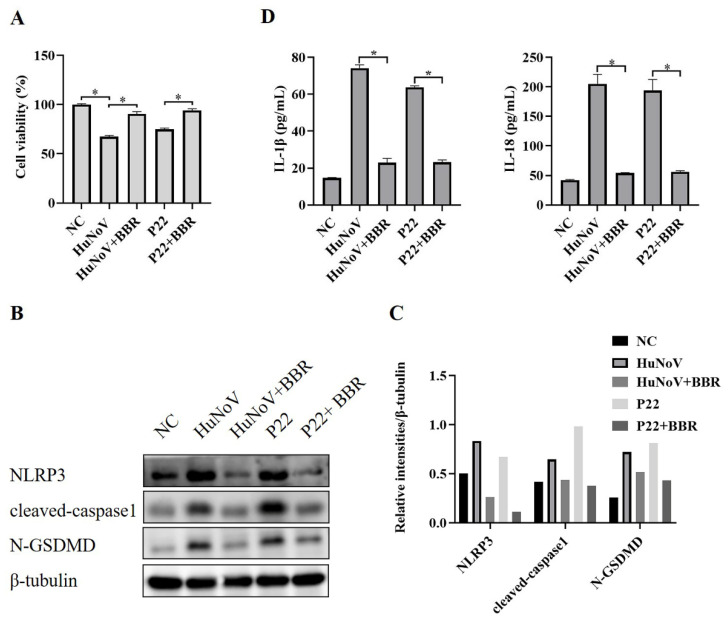
BBR reduces the cytotoxicity of HuNoV and P22 on Caco2 cells by inhibiting NLRP3 inflammasome activation. Caco2 cells were separately transfected with the full-length HuNoV cDNA clone, P22, or empty vector (NC) for 24 h and then treated with BBR (100 μM) for 48 h. (**A**) CCK-8 assay was used to detect cell viability. (**B**,**C**) The expressions of NLRP3, cleaved-caspase1, N-GSDMD, and β-tubulin in cell lysates were analyzed by Western blot, and then these blots were quantified by Image J. (**D**) The levels of secreted IL-1β and IL-18 in cell culture supernatants were collected and measured by ELISA. * *p* < 0.05.

## Data Availability

The primary data used to support the findings of this study are available from the corresponding author upon request.

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
