# Peer review of "HuNoV Non-Structural Protein P22 Induces Maturation of IL-1β and IL-18 and N-GSDMD-Dependent Pyroptosis through Activating NLRP3 Inflammasome"

_vaccines, 2023, doi:10.3390/vaccines11050993_

Round 1
Reviewer 1 Report
Norovirus (NoV) is an enteric non-enveloped virus which is the leading cause of gastroenteritis across all age groups. It has been found that norovirus (MNV) can induce the activation of NLRP3 inflammasome in the host after infection. When the NLRP3 inflammasome is activated, caspase1 is self-cleaved and activated, it can cause the maturation and release of IL-1β and IL-18, and then activate a strong inflammatory response and induce cell death,(pyroptosis).By inhibiting the activation of NF-κB signaling pathway, BBR can reduce the expression of NLRP3, thus inhibiting the initiation and assembly of NLRP3 inflammasome, suggesting that BBR can inhibit the activation of NLRP3 inflammasome.
The author conducted the detection of IL-1β and IL-18 in the HIEs and Caco2 cells transfected with HuNoV plasmid by ELISA test. The results showed that, compared with the control group, the contents of IL-1β and IL-18 in the HIEs and Caco2 cells transfected with full-length HuNoV plasmid were higher. HuNoV was shown to promote the maturation and release of IL-1β and IL-18 after infection. Western Blot was used to detect cell proteins in different transfection groups.The results showed that the cells transfected with full-length HuNoV were higher in NLRP3, caspase1, cleaved caspase1. And Maturation of caspase-1 and secretion of IL-1β and IL-18 were significantly inhibited in caspase1 or NLRP3 specific siRNA groups. Transfection of Caco2 cells with full-length HuNoV cDNA reached a significant increase in the level of N-GSDMD and the number of dead cell. Specific NLRP3 or caspase1 targeted siRNA co-transfection inhibited N-GSDMD levels and cell death.
Using the same experimental method, we found that HuNoV nonstructural protein P22 is a key protein that promotes the activation of NLRP3 inflammasome, induces the maturation and release of pro-inflammatory cytokines IL-1β and IL-18, and induces pyroptosis.
Elisa results showed that P22 significantly increased the secretion of IL-1β and IL-18. WB assay showed that P22 also increased NLRP3 and cleaved-caspase1 expression, and cleaved-Caspase1 protein hydrolysis from GSDMD to N-GSDMD, and P22-specific siRNA was found to inhibit NLRP3 inflammassome activation.
To investigate whether BBR inhibits HuNoV or P22-induced NLRP3 inflammasome activation and N-GSDMD -dependent pyroptosis, Caco2 cells were treated with BBR and then transfected with HuNoV CDNA or P22 plasmid, and the number of living cells in the BBR treated group was higher than that in the untreated group. And cleaved-capase1, N-GSDMD, and IL-1β and IL-1 secretion were significantly lower in the BBR treated group than in the untreated group. The authors believe that this study demonstrates that BBR treatment can inhibit HuNoV and P22-induced NLRP3 inflammasome activation.
The results of this article suggest that BBR may play an important role in Hunov-induced acute gastroenteritis, providing a new therapeutic target for Hunov-induced gastroenteritis.
The writing of this paper is a little disorderly, the test methods are not completely listed, and some experiments are not of practical significance. The demonstration of the results is not comprehensive enough, and there is a mistake in writing.
Major concerns:
1. Line 45-49 and 67-72:It has been demonstrated that norovirus (MNV) can induce activation of NLRP3 inflammatories in the host after infection, resulting in proinflammatory cytokines IL-1β and IL-18, and inducing cell death. Why add more experiments to prove it again?
2. Line 77-81:It has been found that BBR can inhibit the activation of NLRP3 inflammasome, thus inhibiting IL-1β and IL-1 secretion and cell pyrosis. Moreover, the increased secretion of IL-1β and IL-1 and the number of pyrocytes induced by HuNoV infection are also induced by the activation of NLRP3 inflammasome. So, this paper again confirms that BBR can inhibit the activation of NLRP3 inflammasome induced by HuNoV infection. Is there any special significance?
3. The logic of the article writing, it would be better to write the construction of 2.8 HIEs((human intestinal enteroids) at 2.2.
4. Test methods: NO1, Cell transfection process, transfection condition, transfection method (physical or chemical method)
NO2: Because the Elisa test results were written before the WB test results, the Elisa test methods should be written before the WB methods.
5. Line 223:It is mentioned that the synthesis and release of IL-1β and IL-18 require two signals, one to promote their expression and the other to promote their maturation and release by activating the NLRP3 inflammasome. Why not test the first signal?
6. Each of these graphs has a contral group but there's no interpretation of the data in that group.
Major point:
1. Line 97: There are no methods of reversing transcription.
2. Line 99: We don't know the virus titer, and the experimental group didn't tell us how to use the virus and how much.
3. Line 146:The PVDF membrane should be soaked in methanol before use.
4. Line 172: How long did it take to add CCK-8 after transfection, what is CCK-8, and at what concentration was it used?
5. Line 209: How did HIEs assess HuNoV's effect on inflammation-related cytokines?
6. Line 240: There is no D figure in Figure 2.
7. Line 250:Can't write increased cell death, you can write it had pyroptosis.
8. Line 253:N-GSDMD-dependent pyroptosis caused by HuNoV is dependent on the activation of NLRP3 inflammasome or caspase-1.
9. Line 270: Wrong writing on CDNA
10. Line271:The argument was that P22 plays an important role in NLRP3 inflammasome activation and N-GSDMD-driven pyrodeath, but no corresponding graph has been added to prove that P22 increases the number of cell deaths.
Author Response
Dear reviewer,
Thanks for your advice, we have revised the mauscript.
Kind regards

Reviewer 2 Report
The manuscript entitled HuNoV non-structural protein P22 induces maturation of IL-1b and IL-18 and N-GSDMD-dependent pyroptosis through activating NLRP3 inflammasome uses transfected HuNoV and P22 DNA to measure the cellular responses of HIE cells. The experimental design is straightforward, however additional details are needed to understand the experiments performed. In addition, it is strongly recommended that the authors enlist a native English speaker to edit the document as there are many areas where words are used incorrectly or the sentences are unclear as written. There were also some concerns about the lack of specific experimental controls that would allow the authors to interpret the data as it is interpreted in the manuscript. All suggested changes and concerns are listed below.
*There are many instances of unclear wording, improper grammar/punctuation, run on sentences, etc. Many of these have been highlighted in yellow throughout the text. It is recommended that the document be edited by a native English speaker.
*Line 14 and Lines 207-208- it is stated that in vitro culture systems for HuNoV are lacking. As the authors note later in the manuscript there are cells that support replication in vitro (several cell types actually), but replication is not robust. This statement should be revised to reflect this reality.
*Line 59 -CARD should be defined.
*Line 108-113 – additional experimental details should be included so readers are able to replicate these methods. As they are written, that is not possible.
*Line 114-124 – the controls used for these experiments should be included in the methods. They are referred to in the text and figure legends but are almost never defined in those places either.
*Line 122- Was the transfection agent left on cells for 48hrs or were the cells transfected and then incubated for 48 hrs. As with the comment above, additional experimental detail is needed for readers to be able to replicate these experiments.
Line 198- Methods for HuNoV infection of HIEs are missing and should be added. Include controls.
*Figure 1: What is the control that’s used in this experiment? Should also be included in methods section as mentioned above. Without knowing what the controls are, it is difficult to evaluate the authors interpretation of the data/results.
*Line 250-253 - The decrease in si-NLRP3 is obvious, but not with si-caspaseI. Could use these blots and perform densitometry compared to B-tubulin to try and determine if there is a difference?
*Line 258 – si-caspase and siNLRP3 only treated cells should be included to ensure that those treatments are not impacting cell viability.
*Figure 4 - For consistency, the blots that are si-treated should be labeled as si treated blots were labeled in previous figures.
*Figure 5 - What about treatment with BBR alone? Is the compound impacting the cellular processes or interfering with transfected plasmids?

Author Response

(The authors gave the same response as above.)

Reviewer 3 Report
Chen et al. have investigated inflammasome action in HIEs infected with HuNoV. They have followed this investigation with transfection of HuNoV proteins in the Caco2 cell line. The authors conclude that the HuNoV non-structural protein P22 is the key driver of NLRP3 inflammasome activation and pyroptosis. Caco2 is a poor cell line of choice for inflammasome studies. Overall, this reviewer believes that data are unsubstantiated and lack the robustness to make these conclusions. This is inherently a result of flaws in the experimental design and approaches to investigating inflammasome activation.
The major issues with the manuscript which are as follows:
1) The authors do no investigate any of their findings in immune cells. This is particularly important since the inflammasome is a critical cytosolic complex which is largely expressed in immune cells and contributes to innate immune responses against pathogens such as NoV. The standard for the field for investigating NLRP3 inflammasome is in monocytes or macrophages, either derived from mice or humans. There is a B-cell model for HuNoV infection that the authors could have applied
2) The readouts for inflammasome activation are not convincing. For example, in Figure 1 (and several other figures), the level of IL-1b and IL-18 quantified from the supernatant of HuNoV-infected HIEs and HuNoV/p22-transfected Caco2 cells is minimal (<200 pg/mL) and would be considered background in the inflammasome field. Moreover, there is no positive control included in the figure to illustrate the robustness of inflammasome activation and therefore no suitable comparison to judge whether this is truly inflammasome activation or just background noise.
3) Given the lack of confidence in the cytokine readout for Figure 1, additional measures of inflammasome activation would need to be quantified e.g. western blotting for caspase-1 and gasdermin-D and some measure of cell death to quantify (e.g. PI/SYTOX uptake or LDH release). Further, given the authors are claiming NLRP3 activation in HIEs, confocal microscopy analysis of inflammasome markers using this model would have strengthened the findings.
4) In Figures 2 +3, the siRNA knockdown of NLRP3 is subtle at best. The western blot analysis for NLRP3 shows minimal or no difference compared to the siRNA control, and then remarkably there is a significant reduction in cytokine production. The authors should have quantified the protein knockdown to illustrate any differences in expression. Again, no positive control is present to make proper comparisons or assessment of the robustness of inflammasome activation.
5) In Figure 4, it is entirely unclear why the authors have chosen to overexpress a single HuNoV protein (in this case P22) in Caco2 cells to measure inflammasome activation. How do they know that only P22 results in activation of NLPR3 and not others? It is unclear why the authors did not choose to overexpress other HuNoV proteins to illustrate that their measurements are specific to P22. This again illustrates a lack of controls which weakens the dataset.
6) In Figure 5, the authors use berberine (BBR) in attempt to block inflammasome activation. BBR affects the priming signal (step 1 of NLRP3 inflammasome activation) and therefore reduces the expression of NLRP3 and they confirm this using western blotting against NLRP3. It is therefore unsurprising that there would be a reduction in NLRP3 activation when the abundance of this protein is less in the cell.
7) Lastly, overall, the majority of the study is carried out with and overexpression of viral proteins. Naturally, infection with live virus would be much more appropriate. That aside, the authors don’t provide any data on the efficacy of their viral protein overexpression. Again, lacking the suitable controls for such work.
Author Response

(The authors gave the same response as above.)

Round 2
Reviewer 2 Report
The authors made the requested improvements to the manuscript. The English is much improved and the manuscript much clearer. In addition, the inclusion of methods and discussion of controls allows for better evaluation of the data. These changes have dramatically improved the manuscript, but now that data is better analyzed there are some additional revisions needed.
1. Figure 1. Is there any indication if the phenotypes with live virus are linked to replication? Was viral genome measured after incubation? Is the figure accurate that a plasmid was used as the control for purified live virus? If so, that is not a proper control. Stool is a complex matrix and other immune stimulators are isolated alongside virus (i.e. other viruses, host vesicles, bacterial vesicles, etc.). The proper control would be use of norovirus-free stool.
2. Figure 2, line 286. Authors state results are significantly different, but no statistics are done. The best recourse would be to perform densitometry on other blots to demonstrate that the results were reproducible among multiple samples. In that way statistics could be performed. At a minimum, the word “significant” needs to be removed and authors need to comments on how the shown blot compared to other experimental replicates.
3. Figure 3, line 307. Same comment as above. Significant cannot be used.
4. Figure 5. Figure 5C is not mentioned in the text.
Author Response
Reviewer 1 #:
The authors made the requested improvements to the manuscript. The English is much improved and the manuscript much clearer. In addition, the inclusion of methods and discussion of controls allows for better evaluation of the data. These changes have dramatically improved the manuscript, but now that data is better analyzed there are some additional revisions needed.
- Figure 1. Is there any indication if the phenotypes with live virus are linked to replication? Was viral genome measured after incubation? Is the figure accurate that a plasmid was used as the control for purified live virus? If so, that is not a proper control. Stool is a complex matrix and other immune stimulators are isolated alongside virus (i.e. other viruses, host vesicles, bacterial vesicles, etc.). The proper control would be use of norovirus-free stool.
RE:Thanks for your suggestion and we are sorry to make you confused. As described in the 2.1 section, the sequence of HuNoV GII.4 used here was isolated from positive stool samples, and the progeny viruses used in this work were obtained by transfecting HEK293T cells with the plasmid encoding the full-length HuNoV cDNA. The purified virus was kept in a preservation solution for later use. Therefore, in Figure 1A, the virus preservation solution was used in NC group. And we have revised it in the manuscript. Besides, progeny viruses obtained by transfecting HEK293T cells with the plasmid encoding the full-length HuNoV cDNA could successfully infect HIEs, and the viral genome was detected after infection.
- Figure 2, line 286. Authors state results are significantly different, but no statistics are done. The best recourse would be to perform densitometry on other blots to demonstrate that the results were reproducible among multiple samples. In that way statistics could be performed. At a minimum, the word “significant” needs to be removed and authors need to comments on how the shown blot compared to other experimental replicates.
RE:Thanks for your advice, and we have revised it.
- Figure 3, line 307. Same comment as above. Significant cannot be used.
RE:Thanks for your advice, and we have revised it.
- Figure 5. Figure 5C is not mentioned in the text.
RE:Thanks for your advice, and we have revised it.
Reviewer 3 Report
Nil.
Author Response
The major issues with the manuscript which are as follows:
- The authors do no investigate any of their findings in immune cells. This is particularly important since the inflammasome is a critical cytosolic complex which is largely expressed in immune cells and contributes to innate immune responses against pathogens such as NoV. The standard for the field for investigating NLRP3 inflammasome is in monocytes or macrophages, either derived from mice or humans. There is a B-cell model for HuNoV infection that the authors could have applied
RE: Thanks for your advice. The Caco2 cell line is of colorectal origin and differentiates spontaneously into intestinal enterocyte-like cells, which was frequently selected as cell models for investigating the pathogenesis of HuNoV induced bowel diseases (Zhang et al. Human Norovirus Induces Aquaporin 1 Production by Activating NF-κB Signaling Pathway. (2022); Zheng et al. Human Norovirus NTPase Antagonizes Interferon-β Production by Interacting With IkB Kinase ε. (2021)). NLRP3 is also expressed in intestinal epithelial cells, and NLRP3 inflammasome activation plays an important role in the etiology of inflammatory bowel diseases. Pyroptosis of intestinal epithelial cells may be one of causes of diarrhea caused by HuNoV infection, so Caco2 cells were selected as the research model to investigate the pyrotosis caused by HuNoV induced NLRP3 inflammasome activation.
- The readouts for inflammasome activation are not convincing. For example, in Figure 1 (and several other figures), the level of IL-1b and IL-18 quantified from the supernatant of HuNoV-infected HIEs and HuNoV/p22-transfected Caco2 cells is minimal (<200 pg/mL) and would be considered background in the inflammasome field. Moreover, there is no positive control included in the figure to illustrate the robustness of inflammasome activation and therefore no suitable comparison to judge whether this is truly inflammasome activation or just background noise.
RE: Thanks for your suggestion. We repeated the experiment several times and got the same result. The levels of IL-1b and IL-18 in the treatment group were mostly indeed lower than 200 pg/ml, but the differences were statistically significant compared with the control group. In addition, it could be inferred from the levels of the control group that the background noise of this experiment was very low, which might be related to many factors, including the ELISA kit, cell models and experimental methods used. And combined with the results of Western blot, we believe that our data is credible and the NLRP3 inflammasome is truly activated.
- Given the lack of confidence in the cytokine readout for Figure 1, additional measures of inflammasome activation would need to be quantified e.g. western blotting for caspase-1 and gasdermin-D and some measure of cell death to quantify (e.g. PI/SYTOX uptake or LDH release). Further, given the authors are claiming NLRP3 activation in HIEs, confocal microscopy analysis of inflammasome markers using this model would have strengthened the findings.
RE: Thanks for your advice. Clinically discarded human intestinal tissue is scarce, so most of the experiments had been performed on cell models. The HIEs results gave us a hint, and then we subsequently verified it in the cell model.
- In Figures 2 +3, the siRNA knockdown of NLRP3 is subtle at best. The western blot analysis for NLRP3 shows minimal or no difference compared to the siRNA control, and then remarkably there is a significant reduction in cytokine production. The authors should have quantified the protein knockdown to illustrate any differences in expression. Again, no positive control is present to make proper comparisons or assessment of the robustness of inflammasome activation.
RE: Thanks for your advice. We have quantified these blots by Image J, and it shows that the knockdown of NLRP3 is obviously. And we have revised the manuscript.
- In Figure 4, it is entirely unclear why the authors have chosen to overexpress a single HuNoV protein (in this case P22) in Caco2 cells to measure inflammasome activation. How do they know that only P22 results in activation of NLPR3 and not others? It is unclear why the authors did not choose to overexpress other HuNoV proteins to illustrate that their measurements are specific to P22. This again illustrates a lack of controls which weakens the dataset.
RE: Thanks for your suggestion. We also detected the effect of other HuNoV proteins on the secretion of IL-1β and IL18 and screened out that P22 could significantly increase their secretion. Therefore, we further investigated the mechanism of P22 increased secretion of IL-1β and IL18.
- In Figure 5, the authors use berberine (BBR) in attempt to block inflammasome activation. BBR affects the priming signal (step 1 of NLRP3 inflammasome activation) and therefore reduces the expression of NLRP3 and they confirm this using western blotting against NLRP3. It is therefore unsurprising that there would be a reduction in NLRP3 activation when the abundance of this protein is less in the cell.
RE: Thanks for your advice. The objective of this work is to explore the mechanism of NLRP3 inflammasome activation and N-GSDMD dependent pyroptosis induced by HuNoV. BBR could block the activation of NLRP3 inflammasome, but it is not clear whether BBR can block HuNoV induced activation. The purpose of this experiment was to investigate and determine the potential role of BBR in the treatment of HuNoV induced gastroenteritis, the results showed that BBR could alleviate the pyroptosis caused by HuNoV. - Lastly, overall, the majority of the study is carried out with and overexpression of viral proteins. Naturally, infection with live virus would be much more appropriate. That aside, the authors don’t provide any data on the efficacy of their viral protein overexpression. Again, lacking the suitable controls for such work.
RE: Thanks for your advice. Human intestinal enteroids (HIEs) have been successfully constructed and demonstrated to be able to support the replication of HuNoV, but clinically discarded human intestinal tissue is scarce, so Caco2 cells were frequently selected as research models to investigate the pathogenicity mechanism of HuNoV. Here, Caco2 cells were selected to investigate the pyrotosis caused by HuNoV induced NLRP3 inflammasome activation. HuNoV could’t efficiently infect Caco2 cells, so most of our experiments were done by transfection. And P22 was overexpressed, as shown in Figure 4B-C. We have revised the manuscript.